# Polyphenylene Sulfide-Based Membranes: Recent Progress and Future Perspectives

**DOI:** 10.3390/membranes12100924

**Published:** 2022-09-24

**Authors:** Yuan Gao, Xinghai Zhou, Maliang Zhang, Lihua Lyu, Zhenhuan Li

**Affiliations:** 1School of Textile and Material Engineering, Dalian Polytechnic University, Dalian 116034, China; 2State Key Laboratory of Separation Membranes and Membrane Processes, National Center for International Joint Research on Separation Membranes, School of Materials Science and Engineering, Tiangong University, Tianjin 300387, China

**Keywords:** polyphenylene sulfide, membrane, application

## Abstract

As a special engineering plastic, polyphenylene sulfide (PPS) can also be used to prepare membranes for membrane separation processes, adsorption, and catalytic and battery separators because of its unique properties, such as corrosion resistance, and chemical and thermal stability. Nowadays, many researchers have developed various types of PPS membranes, such as the PPS flat membrane, PPS microfiber membrane and PPS hollow fiber membrane, and have even achieved special functional modifications. In this review, the synthesis and modification of PPS resin, the formation of PPS membrane and the research progress of functional modification methods are systematically introduced, and the future perspective of PPS membrane is discussed.

## 1. Introduction

In the last half a century, membrane separation and membrane processes have been applied to all walks of life, which greatly promotes the development of relevant industries and improves the quality of life for people [1,2]. As the core component of membrane modules, membrane material has great influence on the structure and properties of membranes, which can be divided into organic material and inorganic material. The common membrane materials include ceramics, metal oxides and organic polymers. Ceramic materials are generally considered to possess excellent thermal stability and chemical corrosion resistance, but ceramic membranes are greatly restricted due to their low fracture toughness [3]. Polymer materials have opposite properties, and the thermal stability and chemical corrosion resistance are weak, but they possess good toughness and easy processing [4]. Most of the traditional membrane separation fields are concentrated in common wastewater treatment, seawater desalination, oil–water separation, neutral gas separation and so on [5], but there are still many waste solvents, a high concentration of sewage, high temperature tail gas and corrosive solid waste, which need to be treated urgently in the special separation fields of medicine, energy, petrochemical and smelting. In addition, with the rapid development of battery and electrolytic cell industry, the demand for high-performance separator is growing [6]. The electrolyte contains complex chemical components, such as strong polar organic solvent, and acid and alkali solution [7]. In order to keep the battery running stably, the thermal stability of the separator is also required. Therefore, the research and development of high-performance membranes with high temperature resistance, organic solvent resistance, acid/alkali resistance and oxidation resistance has become an important research direction for membrane science and technology and one of the research hotspots in the field of polymer materials for science and engineering.

PPS, as a thermoplastic polymer, is formed by alternating the connection of benzene rings and sulfur atoms, and the structural formula is shown in Figure 1 [8]. Its molecular structure is strong, owing to the existence of benzene rings, and the embedding of sulfur ether bond can endow it a certain degree of flexibility, so the special structure also resolves the excellent performance. PPS resin’s density (*ρ*), glass transition temperature (Tg) and melting temperature (Tm) are 1.34 g/cm^3^, 85 °C and 285 °C, respectively [9,10,11]. PPS is one of the best thermal stabilities of all thermoplastic resins; its decomposition temperature in the air is more than 450 °C, and the long-term service temperature is about 200 °C. It is widely used in environmental protection, the automobile, electronics, machinery, chemical and pharmacy industries, among others. Meanwhile, PPS also possesses the excellent chemical corrosion resistance, and almost no solvent can dissolve it at below 200 °C [12]. In addition to strong oxidizing acid, PPS can resist the corrosion of almost all acid, alkali and high concentration salt solutions [13,14,15]. Compared with other high-performance engineering plastics, PPS has better cost performance, which makes it have the potential of a wide range of separation membrane fields in harsh environments [16,17,18]. Therefore, PPS-based membrane products are very suitable for application in extreme environments, especially in special separation membranes, battery and electrolytic cell separators, which have attracted many researchers for development and application.

In this paper, we mainly focus on the preparation and application of PPS materials and various shapes of membranes for a comprehensive analysis (Figure 2), and we have collected a large amount of relevant literature and reports, such as the research progress of PPS synthesis, preparation and modification methods of PPS flat membrane, ultrafine fiber membrane and hollow fiber membrane. Additionally, the future development perspective and direction of related industries is put forward.

## 2. PPS Resin Synthesis

In 1888, PPS as a by-product of chemical reaction was discovered by accident for the first time, then people began to explore its polymerization mechanism [19,20]. Up to now, many synthetic routes have been developed, such as the sodium sulfide (Na_2_S) method, sulfur method, Genvresse method, Macullum method, Phillips method, oxidative polymerization method, amorphous PPS synthesis method and diphenyl disulfide synthesis. The Phillips method and sulfur method are mainly used to synthesize PPS in industry.

### 2.1. Phillips Method

In 1967, Phillips Petroleum Company [21] applied for a patent for the synthesis of linear PPS resin, and this technology was named Phillips method or sodium sulfide method and successfully realized industrial production in 1973. Using anhydrous Na_2_S and p-dichlorobenzene (p-DCB) as raw materials, a certain amount of alkali metal as promoter and catalyst, PPS with linear structure and high molecular weight was prepared by polycondensation in strong polar organic solvent under high temperature and high pressure. The reaction formula was as follows:

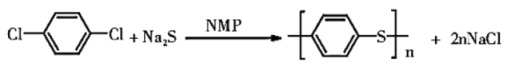


This method had the advantages of stable quality, good repeatability and high yield, which was the most important industrial production method in the world. However, this synthesis method required the high purity of raw materials, and the moisture resistance, electrical properties and molding properties of PPS product decreased due to the trace Na ions in the product and the crosslinking deformation caused by heating. In addition, lithium chloride was usually used as a catalyst for the synthesis reaction, but the price of lithium chloride was increased owing to the development of the lithium–ion battery, causing the increase in production cost.

### 2.2. Sulfur Method

Sulfur method was a unique production method of polyphenylene sulfide in China, and the industrial trial production had been realized in related enterprises. In 1988, Yongrong Chen et al. proposed for the first time that dichlorobenzene and sulfur as raw materials were polycondensated to produce PPS resin at 175–250 °C under normal pressure in polar solvents, such as hexamethylphosphoryltriamide or N-methylpyrrolidone [22]. The reaction formula was as follows:

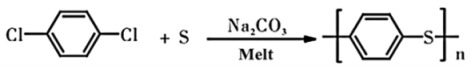


This preparation method had many advantages, such as the high product yield (over 85%), great product quality and short reaction cycle. However, the sulfur purification technology was difficult, and it was easy to introduce impurities in reaction system. In the process of industrial production, the refining and purification of sulfur was also one of the technical difficulties; the amount of by-product waste salt was significant.

### 2.3. Other Methods

In 1897, Friedel crafts catalyst was used to catalyze the polymerization reaction of benzene and sulfur to produce PPS under heating, but the yield and molecular weight were very low, the molecular branching degree and crosslinking degree were high and it was not easy to dissolve [23]. In 1910, Hildtich et al. [24] found that thiophenol could self-polymerize to produce amorphous PPS powder in concentrated sulfuric acid solution. In 1948, p-dichlorobenzene, sulfur and sodium carbonate were used as reactants to carry out pressurized solid-state melt polycondensation at 275–360 °C [25]. The obtained PPS resin was provided with the great chemical stability and mechanical properties, but the low molecular weight caused it easy disproportionation and crosslinking, and the chain was broken easily at high temperature, resulting in poor thermal stability. In 1989, Lewis acid catalyst was used to catalyze the oxidative polymerization of diphenyl disulfide to produce PPS resin at room temperature and atmospheric pressure. This method was provided with the low cost and high yield at 100%, but the molecular weight of product was not high, so it had no practical application value [26].

### 2.4. Synthesis Methods of Resins with Special Structure

Based on the previous research results of PPS resin synthesis, the researchers also developed various PPS resins with different structures, which could meet the needs of various special application fields.

#### 2.4.1. Linear High-Molecular Weight Resin

Due to its excellent performance, PPS resin with a high-molecular weight has become the research focus in various countries, mainly including linear and branched chain. Owing to the great fluidity, linear resin could be directly processed and used, which had attracted extensive attention. H_2_S, NaOH and p-DCB were used as raw materials and anhydrous CH_3_COONa and Na_2_CO_3_ as composite catalyst to synthesize PPS resin with high-molecular weight at 270 °C and 490–980 KPa [27,28]. However, harsh reaction conditions (high pressure and high temperature), long reaction time and difficult post-treatment processes resulted in low PPS yield. H_2_S as raw materials was easy to obtain, but the strong corrosive led to high performance requirements for the equipment, limiting the industrial application of this method. Jixing Luo et al. [29] developed a method for preparing linear high-molecular weight resin by atmospheric pressure polycondensation with refined H_2_S, NaOH and p-DCB as raw materials and alkali metal salts as additives in hexamethyl phosphoryl triamine (HPTA) solvent system. There were also a few side reactions in this reaction process, and the product with high linearity and high quality was obtained. However, the generated waste gas could cause serious pollution, and the post-treatment of waste gas was complex, which limited the wide application of the technology, especially the global sustainable development strategy. Multi-component catalyst and refined industrial sodium sulfide were used as a sulfur source to synthesize linear high-molecular weight resin by segmented polycondensation in HMPA system, which avoided the dehydration problem of sodium sulfide, thus providing the great industrial prospects [30]. Wenwei Jiang et al. [31] synthesized the linear high-molecular weight resin by using lithium sulfide as a reaction monomer, which possesses the similar synthesis mechanism as the sodium sulfide method. Compared with the traditional sodium sulfide method, the obtained PPS resin synthesized by this method possessed the linear molecular structure, higher molecular weight and better thermal stability.

#### 2.4.2. Branched High Molecular Weight Resin

The PPS resin with branched chain has poor fluidity, difficult processing and low crystallinity, so it was suitable for plastics and laminated materials. Campbell [32] and Edmonds [33] proposed that the branched chain PPS resin with high molecular weight at 200,000 could be synthesized under high pressure in polar organic solvent N-methylpyrrolidone (NMP) with alkali metal sulfide and p-dichlorobenzene as raw materials, active 1,2,4-trichlorobenzene as a third monomer and alkali metal salt as additives. The reaction formula was as follows:

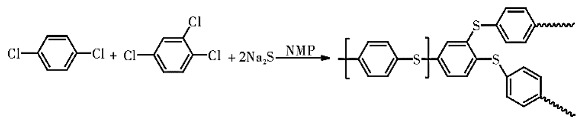


Sulfur and p-dichlorobenzene were also used as monomers and added 2,5-dichloronitrobenzene with reactive group -NO_2_ as comonomer for polymerization. In this reaction, -NO_2_ was reduced to -NH_2_, and branched high molecular weight resin was obtained via further branching with aromatic ring [34]. Methyl 3,5-diphenylthiosulfoxide as monomer and 1,3,5-tribromobenzene were also used as molecular weight regulator to synthesize 1-methylmercapto-3,5-diphenylthiobenzene via cationic polymerization. Then, the monomer was oxidized by nitric acid, and forming hyperbranched PPS by self-polycondensation in CF_3_SO_3_H [35].

#### 2.4.3. Resin with Low Melting Temperature

Owing to the melting point and processing temperature of PPS resin being above 285 °C and 300 °C, respectively, it was difficult to process and apply to industrial production. Therefore, it was important to develop PPS resin with low melting point to reduce processing difficulty and expand the application field of resin. Yongrong Chen et al. [36] used p-dichlorobenzene and anhydrous sodium sulfide as a reaction monomer, dimethyl sulfoxide as solvent and a small amount of crown ether or polyethylene glycol as additives to obtain low molecular weight PPS resin with a melting point of 122–195 °C. The obtained PPS resin could be oxidized and crosslinked after heating treatment, which was suitable as heat-resistant and anti-corrosion coating materials. The linear molecular structure could reduce the melting point of the PPS resin with high molecular weight. Huiyong An et al. used sulfur and dichlorobenzene as monomers, lithium chloride and sodium phosphate as a catalyst to synthesize PPS resin by self-polycondensation reaction in a hexamethylphosphoramide (HMPA) solvent system. Additionally, the melting point reached 240 °C, and its average molecular weight reached more than 60,000 [37]. Zhenhuan Li et al. developed the high-performance PPS resin synthesis technology for synthetic fiber, and membrane and melt-blown grades in the non-lithium chloride system, which had realized the various PPS industrial production in Tianjin Petrochemical Company [38]. These research results greatly improved the cost performance of PPS resin, and promoted the large-scale application of PPS resin in other industrial fields.

At present, there are three main problems in the synthesis of high-performance PPS resin: (1) Purification of raw materials. The purity of sodium sulfide is poor, and the impurities, such as 1-chlorobenzene, o-dichlorobenzene, M-dichlorobenzene and Trichlorobenzene, in p-dichlorobenzene will influence the purity of resin; (2) Side reactions of solvents in synthetic system. In the initial stage of the reaction, NaOH generated from the reaction of water molecules and auxiliaries will lead to the ring opening of solvent NMP, thus affecting the further increase in molecular weight of PPS resin; (3) Purification of PPS product. A lot of low molecular oligomers and sodium salts will be formed in reaction process, which will cause the resin to possess strong hygroscopicity, poor corrosion resistance and thermal stability.

## 3. PPS Flat Membrane

PPS flat membranes were prepared via membrane scraping process and modification process, and have been successfully applied in various membrane separation processes, including oil–water mixture separation, membrane distillation, seawater desalination and removal of small molecules in an aqueous and solvent system.

### 3.1. PPS Flat Membrane Preparation Process

Due to the special properties of PPS resin, that is, almost no solvent could dissolve it at under 200 °C, PPS membrane was usually prepared by the thermally induced phase separation (TIPS) method. The detailed process is listed as follows. The organic solvents with high boiling point and similar solubility parameters were selected as diluents to dissolve PPS resin at high temperature, and homogeneous casting membrane solution was obtained. Then, the PPS liquid membrane was prepared by scraping, pressing and spinning method, followed by the phase separation process at low temperature and extraction process obtaining the PPS membrane. In 2006, Ding et al. [39,40] selected diphenyl sulfone and diphenyl ketone (DPK) as diluents to prepare PPS porous polymer material with excellent solvent resistance and heat resistance by TIPS method, and the formation of pore structure was the result of competition mechanism between L-L phase separation and polymer crystallization. In detail, the nucleation growth of polymer-rich phase in the L-L phase-separation process and the crystallization of polymer in the polymer-rich phase decided on the pore structure. On the basis of these theories, there were several studies on the preparation of PPS flat membrane by TIPS method, which gradually acquired the valuable PPS microfiltration membrane. Wang et al. [41] also chose the mixed diluents and adjusted the ratio to prepare a casting membrane solution with the cloud point and coarsening time, which could affect the phase separation process and coarsening process. This results showed that the pore structure, pore size and branch thickness of membrane prepared by mixed diluents was different from that prepared by single diluent. The nucleating agents played one of the important roles in the forming of porous structure via TIPS method. In order to obtain the better pore structure and higher porosity, polyethylene glycol with different molecular weight and different concentration as nucleating agent could change the pore structure from spherical structure to the compact branch-like structure [42], so nucleating agent as the second phase could increase the nucleation density and then decrease the size of spherulites. Thus, for the phase separation mechanism, the addition of nucleating agent could change the surface energy of spherulites, which made the phase separation process more rapid and without spherical structure formation.

Previous studies mainly focused on the pore structure formation mechanism of PPS porous materials, but they did not describe the detailed membrane-forming equipment and process. It was only in recent years that PPS membrane had really appeared in the laboratory. Zheng et al. [43] was the first to design the new casting device shown in Figure 3, including electrical heating plate, aluminum film and casting bar, which was also commonly applicable to the flat membranes preparation of polyethylene (PE), polyvinylidene fluoride (PVDF), isotactic polypropylene (PP) and other materials [44,45,46].

The phase separation process was established by thermal analysis of solidified casting solution, including liquid–liquid phase separation behavior in DPS, DPIP and benzoin (BZ) and solid–liquid phase separation behavior in CPL, HTP and CHPN. Via the same casting membrane device, and the diluent and auxiliary diluent were introduced as the compound diluents to control the membrane-forming process [47]. The auxiliary diluent could decrease the compatibility between polymer and mixed diluents, which inhibited the movement of the PPS molecular chain by increasing the viscosity, causing the phase separation process to be changed from S-L phase separation to L-L phase separation and forming a more ideal membrane structure with asymmetric structure, as shown in Figure 4. This process was the most widely used method to prepare PPS microporous membrane, and it had been developed into various functional composite membranes via various modification processes. Subsequently, the super-hydrophobic PPS microporous membrane with Berberis thunbergii var. atropurpurea Chenault leaf-like structure was prepared via DPK and BZ as compound diluents, which was successfully used in the vacuum membrane distillation (VMD) process [48]. The phase separation process was adjusted by controlling the ratio of DPK content, and it was found that the membrane structure changed from honeycomb structure to bi-continuous structure and spherulite structure. 

To date, PPS membranes with the strong chemical corrosion resistance, solvent resistance and high temperature resistance have been successfully prepared in the laboratory, which can be used in various fields. However, due to the slow technical progress in the synthesis of PPS resin, it has not been able to obtain the enough high molecular weight resin comparable to conventional membrane materials, such as PVDF, PAN, PSF and PES. In the membrane preparation process, the viscosity of the casting solution is too low, resulting in poor membrane formation. Researchers generally solved this problem by adding non-solvent component additives, but the prepared membranes showed poor toughness and low strength. Therefore, another good method is to increase the molecular weight. Lundgard et al. used PPS resin after oxygen treatment as the membrane material, which can improve the viscosity of the casting solution in the membrane preparation process to improve the membrane-forming property [49]. In addition, the instability of high temperature TIPS process leads to the poor uniformity of membrane structure and low surface pore density. Therefore, researchers still need to spend substantial efforts in solving these problems to obtain excellent PPS membranes.

### 3.2. PPS Flat Membrane Modification Process

In order to expand the application field of separation membrane, researchers developed a variety of methods to modify the membrane and membrane materials. One route was to prepare mixed matrix membrane by adding modified materials into the casting membrane solution; and another route was prepared surface functionalized membrane by post-treatment modification [50,51,52].

The physical blending method was one of the simplest and most commonly used modification methods, which was used to uniformly disperse different polymer or inorganic particles. Additives and polymers gave play to their advantages, and skillfully made up for their respective performance defects, to achieve a win–win situation [53]. Based on the characteristics of PPS resin and membrane, researchers developed a variety of modification methods to improve the overall membrane performance. PPS resin with poor UV-resistance properties could manifest the membrane in the yellowing and darkening of color, and in a serious reduction of the mechanical properties under the effect of UV photodegradation [54,55]. As shown in Figure 5A, TiO_2_ nanoparticles modified with SiO_2_ and 3-aminopropyltriethoxysilane (APTES) were added into the casting solution, and the prepared PPS composite membrane with excellent anti-UV degradation performance [56]. Comparing the parental PPS membrane with PPS composite membrane, the anti-UV degradation performance was significantly enhanced, which was owing to the lower photocatalytic activities, higher UV-shielding abilities and easy dispersion in PPS matrix. The benzene ring of the PPS molecular structure determined its inherent hydrophobic characteristics [57,58], so it had potential to develop a super-hydrophobic PPS membrane via self-construction or introduction of a superhydrophobic modifier for treating oil–water emulsion. Hydrophobic SiO_2_ nanoparticles were added as additive into the PPS/BZ/DPK system forming a homogeneous casting solution, and the superhydrophobic mixed matrix PPS membrane with a lotus leaf-like micro–nano structure were prepared [59]. Hydrophobic SiO_2_ nanoparticle distributed on the membrane surface and pore wall played an important role in the membrane properties. As shown in Figure 5B, the obtained membrane was successfully used in the water in oil separation process, and owing to the inherent characteristics of PPS resin, it could be used in the oil–water separation process in a strong acid and alkali environment. Carbon materials have been widely used as modified additives in mixed matrix membranes, such as graphene and carbon nanotubes. Porous carbon nanofiber (PCNF) was used as a modifier to prepare PPS/PCNF composite membrane in one study [60], and PCNF was able to migrate to the top surface of the composite membrane and form a new functional layer. Compared with neat PPS membrane, PPS/PCNF composite membrane with higher solid content was provided with a smaller pore-size and narrower pore-size distribution. Meanwhile, the nanofibers were combined with the molecular chains by a physical–chemical effect causing the enhancement of the mechanical properties. Although the physical blending modification process was simple and the mechanism was clear, almost all of the obtained composite membranes had inhomogeneous blending of components, which led to various problems during use. For example, the defects caused by the accumulation of modifiers could lead to the decrease in selective permeability and mechanical properties of membranes. Therefore, the focus of this kind of process was always how to blend evenly. It was more effective to improve the compatibility of the additives and the components of the casting solution, with the exception of stirring violently and ultrasonic dispersing [61]. Wang et al. selected CPL as solvent, TiO_2_ and graphene oxide (GO) as modification additives to prepare a PPS/TiO_2_/GO membrane with better water permeability and anti-pollution performance. In this casting membrane system, CPL could form hydrogen bonds with the oxygen-containing functional groups of TiO_2_ and GO, so the dispersion of nanoparticles in the casting membrane solution was effectively improved [62]. Meanwhile, the addition of hydrophilic nanoparticles enlarged the separation degree of the liquid–liquid phase, and the coarsening time was longer thereby causing the formation of a smart continuous network structure with higher connectivity.

The membrane surface modification was to change the chemical composition of membrane surface by physical and chemical methods, and then endowed the membrane with new functionality. This method could effectively improve the selectivity, permeability and stability of the polymer membrane without destroying its main support structure. At present, the common methods of membrane surface modification included surface physical modification and surface chemical modification. Surface chemical modification could effectively solve the problem of modified layer dropping by chemical bonding and improve the running stability of membranes. PPS flat membrane was modified with low concentration nitric acid solution, because of its poor oxidation resistance, obtaining superhydrophilic PPS membrane with a large number of hydrophilic groups, such as -SO-, -SO_2_-, C=O, -NH_2_ and -NO_2_ groups. The modification process was simple, and low concentration nitric acid could only modify the membrane surface without damaging the membrane matrix. However, the reaction mechanism was complex, including oxidation reaction, weak-degradation reaction and nitration reaction [51]. In order to solve the inherent poor antibacterial property and weak anti-protein contamination ability of PPS membrane, hydrophilic and anti-biofouling PPS microporous membrane was developed via chloromethylation and quaternary amination reaction [63]. The introduction of ammonium salts and imidazolium salts effectively caused the antibacterial activity improvement of modified membranes.

Surface physical modification was to endow the membrane with some functional properties by surface coating or surface adsorption [64], these methods were simple and efficient. It could not only retain the mechanical properties of based membrane, but also obtain the properties of the modified layer, and the main functional properties came from the modified layer. Dopamine (DA) was known as Mussel-inspired “biological glue” and had been used as a modified agent in membrane modification, which showed the excellent adhesion strength with materials surfaces [65,66,67]. DA and polyethyleneimine (PEI) were used to construct functional layer on the surface of PPS membrane via oxidative self-polymerization and co-deposition, which showed the excellent separation performance for dyes wastewater and service durability [68]. The method possessed the characteristics of simple operation and universality, showing the same modification effect on different membranes materials. Because the PDA layer contained a large number of amino groups, it was convenient for the subsequent modification process, which could greatly expand the application field of the PPS membrane. A rough TiO_2_ layer was constructed on the surface of PPS membrane after nitric acid treatment via electrostatic assembly, which is shown in Figure 6. A superhydrophilic TiO_2_@h-PPS membrane was prepared and successfully applied to oil in water emulsion treatment with strong long-term stability [69,70]. Soon afterwards, they also used 1H, 1H, 2H and 2H-perfluorodecyltriethoxysilane (PFDS) as a grafted modifier and TiO_2_@h-PPS on the oil–water interface as based membrane to prepare Janus membrane with one-sided superhydrophobic functional layer and another one-sided superhydrophilic functional layer, which showed a highest water contact-angle difference of −150° between its two surfaces. Janus membrane was a very popular membrane formation, which could greatly expand the application field of the membrane, and it could use the same membrane treating different systems with completely different properties [71,72]. They also used the obtained TiO_2_@h-PPS membrane for photodegradation of organic pollutants. The coupling effect of TiO_2_ and conjugated polymer membrane could improve the photocatalytic activity and photocatalytic efficiency of TiO_2_ [73]. This was due to the strong interaction between TiO_2_ and the membrane matrix, which led to the effective migration and transfer of electron–hole pairs. Meanwhile, the adsorption of PPS on organic pollutants could effectively enhance the catalytic degradation ability of TiO_2_. This process was also the first time that the conjugated structure of PPS materials was used to enhance optical properties, which was of great significance to the development of PPS membrane catalytic process. However, worthy of the researchers’ attention was the question: how to avoid the degradation of PPS membrane in the process of photocatalytic degradation leading to the degradation of mechanical properties?

The PPS membranes prepared by TIPS method and PPS modified membranes prepared via various methods mentioned above all belonged to the category of microfiltration membranes, which were powerless for the interception of small organic molecules and metal salt ions [74]. However, the preparation of PPS nanofiltration membrane could not be realized by the existing technology; constructing nanofiltration functional layer on PPS membrane surface was a good choice. Graphene oxide (GO), as a derivative of graphene, is a kind of two-dimensional nanostructure with a dominant sp^2^ carbon hybridization [75,76,77], and has been used considered as ideal material to prepare the high-performance separation membranes [78,79]. GO membrane has become a unique research direction and relies on the interlayer selective permeation transport of GO interlayers to realize the separation of a variety of mixed systems; thus, researchers have been constantly modifying its interlayer physical and chemical structure of GO layer to achieve a variety of separate objects. In one study, PPS microporous membrane was selected as based membrane, and GO sheets modified by Bis-(triethoxysilyl) ethane (BTESE) were used to construct a nanofiltration functional layer on the surface by vacuum assisted self-assembly method [80]. The introduction of hydrophilic PB nanoparticles not only increased the free-space size between GO sheets, but also formed continuous hydrophilic transport channels between GO sheets, which could effectively improve the permeability of water molecules and ethanol molecules. Until now, the main purpose of GO membranes research has been to improve the permeability of small molecules while retaining the separation efficiency. The main reason limiting the permeability of GO membrane was the long and tortuous permeation channel among GO interlayers, then researchers designed the structure with a metal–NGO framework on the surface and in the pore channels of the PPS membrane support [81]. In these works, GO or its derivatives was fixed on the surface of PPS-based membrane by Van Der Waals force, such as electrostatic adsorption and hydrogen bonding, causing easily falling off. Simultaneously, it was also a common fault of all GO membranes. In order to enhance the binding force between GO layer and based membrane, most researchers advocated using chemical bond to crosslink and fix. Thus, the PDA/PEI co-deposition modified PPS membrane was used as based membrane, and they constructed NGO/PA hybrid layer with disordered stacking structure through vacuum-assisted self-assembly technology and interfacial polymerization [82]. The PDA/PEI co-deposition layer could effectively improve the stability of the hybrid layer by chemical bond fixation. The obtained functional layer possessed disordered arrangement and dual-channel structures, which caused the improvement in selectivity and permeability. However, it unfortunately could not be used in the nanofiltration membrane process under harsh environments, such as high-temperature nanofiltration and organic solvent nanofiltration; therefore, our next research focused on the development of PPS nanofiltration membranes for extreme environments.

### 3.3. Development Prospects of PPS Flat Membrane

Because of its unique properties, PPS resin was very suitable for the membrane separation process, and flat membrane was the basic membrane formation, so the development of PPS flat membrane has important practical application and research significance. Almost three decades after its discovery, PPS flat membrane was developed into a variety of membrane separation processes, such as microfiltration, ultrafiltration, nanofiltration membrane, and had been successfully used in many fields, such as oil–water separation, seawater desalination and dye wastewater treatment. Our research on PPS membranes not only focuses on the construction of sieve-pores, but we also studied the adsorption of pollutants and the synergistic catalytic degradation with metal oxides. Despite current achievements, many issues remain unresolved, and the future development of PPS flat membrane should focus on the following points: (1) By controlling the phase separation process or developing other membrane-forming technology to exploit PPS nanofiltration membrane for organic solvent nanofiltration systems; (2) Combined with the special structure of PPS molecule to develop PPS composite catalytic membrane, explore the catalytic mechanism, and use it in the catalytic system in special fields; (3) To improve the mechanical properties of PPS flat membrane by various methods. Up to now, the common failing of poor mechanical properties has not been solved, which seriously affects the practical application of PPS flat membrane and restricts its industrial development; (4) To develop special separation membrane process. The membrane developed for special purification and separation application in industrial production has strong professional performance, which can be targeted to solve some industrial problems in extreme environments.

## 4. PPS Hollow Fiber Membrane

In the practical application process, the flat membrane with low module loading density is difficult to operate, which cannot carry out high pressure operation, so many shortcomings limit its large-scale application. Compared with the flat membrane, hollow fiber membrane is one of the most widely used membrane forms in industrial production and application, which has the following advantages [83,84,85]: (1) High filling density per unit area. The module has high water production per unit area and high work efficiency; (2) Self-supporting membrane. Hollow fiber membrane does not need support body in operation process, causing simpler process and less material cost; (3) The miniaturization of equipment. Due to the high filling density per unit area of hollow fiber membrane, it is easy to be made into miniaturized and portable equipment for application. Therefore, hollow fiber membrane is one of the most promising membrane forms, and the industrialization of hollow fiber membrane has had a great role in promoting industrial development. 

Due to the special properties of PPS resin and the advantages of hollow fiber membrane forms, the development of PPS hollow fiber membrane has great application prospect. Since the 1970s, researchers in Japan have led the research on the preparation of PPS hollow fiber membrane. Since the 1980s, researchers in European and American countries have also used PPS as membrane material to prepare composite membrane and develop gas separation membrane, which is suitable for a special separation system. Early PPS hollow fiber membrane was prepared via a high temperature melt-spinning, hot-stretching and setting method in a study [86]. Through water flux and the scanning electron microscope (SEM) test, it was found that there were through pores, but low porosity and low surface pore density led to poor water permeability. PPS resin is a semi-crystalline polymer, which meets the requirements of the hole-forming process by stretching method. The membrane material has certain crystallinity requirements, and the process does not need solvent. Under the action of certain temperature and tensile stress, the amorphous region between polymer crystals will produce membrane cracks parallel to or perpendicular to the tensile direction, forming microporous structure. This process is simple to operate, and is also economic, provides environmental protection and is suitable for industrial production. However, this method requires a high molecular weight resin, and there is not this high specification for PPS resin in the current market, which meets this demand. The composite pore-forming agent and supercritical CO_2_ were further introduced mixed with PPS resin to melt spinning, which could effectively improve the porosity and surface pore density [62]. However, the nonuniform dispersion of pore-forming agent in PPS resin melt could lead to defects in the fiber body, causing fiber breakage during the stretching process. Li et al. used BZ and DPK as composite diluent to prepare PPS hollow fiber membrane via TIPS method, and inorganic nanoparticles and nanometer metal salt were added as pore forming agents to improve the opening rate. The obtained hollow fiber membrane possessed higher surface pore-density and porosity, which had a significant effect on the permeability optimization. However, due to the low molecular weight of PPS resin, the high solid content of the casting solution affected the internal structure, including mainly honeycomb pores and only a small amount of double-continuous structure, resulting in the existence of few through channels, which seriously affected the permeability.

At present, despite current achievements, a PPS hollow fiber membrane has not been prepared which can match the performance of traditional commercial hollow fiber membrane. The main reason is the insoluble and high melting point of PPS resin, and the current preparation method is more traditional, which may not be suitable for the preparation of PPS hollow fiber membrane. In order to solve the current dilemma, it is necessary to optimize the raw material synthesis technology and improve the membrane preparation method at the same time. Specifically, researchers should explore the following directions: (1) Developing and synthesizing PPS resin with high molecular weight and narrow molecular weight distribution; (2) Exploring new pore-forming and phase separation mechanisms, such as template, extraction, defects and others; (3) Under the condition of ensuring the mechanical properties as much as possible, the solid content of casting solution can be reduced, which can effectively improve the porosity and through-pore content.

## 5. PPS Ultrafine Fiber Membrane

### 5.1. PPS Ultrafine Fiber Membrane Preparation Process

The fiber with diameter less than 5 μm is called ultrafine fiber, also known as micro-denier fiber, microfiber or fine fiber, which possesses the excellent flexibility, heat preservation, water repellency, air permeability, high specific surface area, strong adsorption, excellent filtration performance, strong capillary effect and so on [87,88,89]. It has been widely used in the traditional clothing industry, filter equipment, biomedical, battery and other fields [90]. The preparation methods mainly include melt-blown, electrospinning, flash evaporation and centrifugal spinning [16,91,92,93,94]. 

PPS ultrafine fiber was mainly prepared by melt-blown method owing to the unique properties of PPS resin. The structure diagram of melt-blown tester is shown in Figure 7, and the product is called PPS melt-blown nonwoven cloth. The experimental system consisted of feeding system, screw extrusion conveying system, melt-blown spray head, air compression and heating system, net belt receiving system and trimming and winding system. PPS resin could be oxidized and crosslinked at a high processing temperature of melt-blown process, and some PPS resin could decompose to produce small molecular gas and low molecular polymer. During the extrusion process of molten PPS resin, it was easy to form globular droplets to block the spinneret hole, resulting in production difficulties, also affecting the fiber fineness and product quality. In order to solve this serious problem, researchers carried out many measures, such as twice filtering process of molten PPS resin, frequent switching on and off of melt blowing equipment and periodic cleaning of melt-blown equipment with PP. However, these methods were not sustainable, which would undoubtedly lead to the decrease in production efficiency and the increase in production costs [95]. PP resin with wide range of melting temperature and large melt viscosity was added into PPS spinning melt to improve the spinnability of PPS resin, finding the introduction of 5% PP resin could improve the PPS/PP composite melt viscosity and alleviate the broken wire phenomenon, which was feasible to improve the spinnability and product quality of PPS melt-blown nonwoven cloth [96]. Phosphite compound with similar effects could also be added into PPS melt to eliminate the polymer particles accumulated in the spinneret hole, obtaining the melt-blown products without defects [97]. Then, PPS melt-blown nonwoven cloth was treated by spun-lace and hot-rolling process to obtain PPS ultrafine fiber membrane. The influence of hot rolling pressure and temperature were also explored on the properties of PPS ultrafine fiber membrane, and the tensile properties of PPS ultrafine fiber membrane could be improved by increasing the hot rolling pressure or temperature [98]. Combined with the strong chemical resistance and thermal stability of PPS resin, PPS ultrafine fiber membrane was surprisingly suitable for special separation, adsorption, battery and electrolytic cell separator.

### 5.2. Application and Modification of PPS Ultrafine Fiber Membrane

#### 5.2.1. Separation Field

PPS ultrafine fiber membrane, even PPS melt-blown nonwoven cloth, possessed a pore size at more than several microns, which was suitable for the interception of ultra-large molecules and large particles, such as the suspended particulate matter (PM) pollution produced by fossil fuel combustion [99,100]. PPS melt-blown fiber net was overlain onto the surface of PPS short fiber net, and adopted array embedding method to reinforce it and form PPS composite filter material, then the composite filter material was heat-set and surface polished. The composite filter material contained a melt-blown micro fiber layer which was firmly bonded. The filter material maintained great air permeability and significantly improved the filtration rate of micro particles, and the preparation method of this filter material was simple and energy-saving. The solar water evaporation system as an emerging solar thermal technology has been widely used in purified drinking water project owing to its characteristics of energy conservation and environmental protection [101,102,103]. A double-layer evaporator with multi-walled carbon nanotubes (MWCNTs) layer and PPS/cellulose (PPS/FC) fiber membrane was prepared by vacuum filtration technology [104]. MWCNT layers showed a high solar absorption rate (about 93%) in the wavelength range of 400–1200 nm and great photo thermal conversion ability, and the PPS/FC membrane with porous network structure in the bottom layer possessed excellent water transport capacity, high temperature stability and great thermal insulation performance. 

The existence of a benzene ring in PPS molecular determined the lipophilic and hydrophobic properties of PPS resin, which was extremely suitable for oil–water emulsion separation. PPS melt-blown nonwovens with entangled fiber structure and internal three-dimensional network structure possessed outstanding oil adsorption capacity, great oil holding performance, fast oil absorption rate and excellent reusability [105]. However, there were oil absorption and de-oiling processes in the using process of oil absorption materials, which led to cumbersome operation and poor work efficiency. Therefore, it is necessary to develop oil–water separation membrane which could work continuously. The mixed polyethylene wax (PEW) powder and polytetrafluoroethylene (PTFE) ultrafine powder were introduced onto the PPS ultrafine fiber membrane surface via one-step spraying method, obtaining superhydrophobic and superlipophilic PPS ultrafine fiber composite membrane [106]. Figure 8 shows the microstructure of coated PPS ultrafine fiber membranes with superhydrophobic and underwater superoil-wet properties, which showed the excellent treatment ability of oil–water mixture. In addition, this simple method was conducive to the large-scale preparation of composite membranes which possessed a broad application prospect in actual oil–water separation. Graphene was a typical hydrophobic carbon material, and graphene coated PPS fiber membrane possessed excellent chemical resistance and hydrophobicity. It could use Joule heating and solar heating to efficiently separate oil–water mixture and quickly adsorb crude oil in all weather, and this work was conducive to the rapid separation of oil–water mixture in all-weather [107]. In the above studies, PPS melt-blown cloth and its modified materials were used for oil–water separation, and the fiber membrane prepared by electrospinning technology possessed the advantages of smaller pore-size and narrower distribution. Therefore, PPS particles were dispersed in PVA solution to prepare PPS composite nanofiber membrane by electrospinning and sintering technology. The prepared composite membrane had excellent hydrophobic properties and chemical stability, which showed good development potential in the field of oil/water purification [108].

#### 5.2.2. Adsorption Field

Excellent chemical resistance, high specific surface area and super-large porosity, as the unique advantage of ultrafine fiber membrane, were found to endow it with great application potential for catalysis, adsorption and degradation [109]. 

Heavy metal wastewater pollution seriously affects the survival of human survival [110,111]; hence, the development of heavy metal wastewater treatment technology, such as adsorption, rejection and conversion, is crucial for people’s health. PPS ultrafine fiber supports zero-valent iron (ZVI) prepared by liquid phase reduction method. It was found that ZVI was uniformly loaded on the fiber surface and the prepared fiber possessed great adsorption efficiency for hexavalent chromium (Cr^6+^) [112]. The processing mechanism was redox reaction, and the adsorption process followed the pseudo-second-order kinetic model. Metal organic frameworks (MOFs) with large specific surface areas and nanocavity structures were widely considered to be very suitable for adsorption, catalytic separation and other fields [113,114,115], and the novel PPS ultrafine fiber composite membranes modified with ZIF-8 and ZIF-8-BSA were prepared via hydrothermal and biomimetic mineralization approaches. Figure 9 described the adsorption mechanism and adsorption capacity of the composite membrane for iodine vapor, showing that the composite membranes could capture trace iodine vapor via ZIF-8 nanocages and surface sites [116]. In addition, it was reported in the literature that PPS materials with rich π electrons showed excellent gas adsorption capability, thus, iodine vapor could also be captured by the phenyl groups of PPS materials via π···I bonding [116].

#### 5.2.3. Lithium Ion Battery and Electrolytic Cell Separator Field

In the battery and electrolytic cell components, the separator existing between cathode and anode was an important role in preventing electron transfer and ensuring smooth ion transmission [117]. PPS resin with the properties of outstanding thermal resistant, inherent flame retardancy, and great chemical stability could be used to prepared lithium ion battery (LIB) and electrolytic cell separator. At present, researchers mainly used surface modification and coating modifier to achieve this goal. For example, PPS ultrafine fiber membrane was used as the support, PVDF and SiO_2_ nanoparticles were selected as surface coating materials to construct high-temperature composite battery separator. Both the relatively high polarity of PVDF and SiO_2_ coating materials and the highly continuous pore structure of the composite membrane could promote the capillary immersion of electrolyte into the membrane micropore structure and rapid diffusion [118]. Compared with commercial membrane PP/PE/PP, the prepared composite membrane possessed high ionic conductivity, high discharge specific capacity and strong heat resistance. As shown in Figure 10, polyvinylidene fluoride hexafluoropropylene (PVDF-HFP) and SiO_2_ were also coated on the surface of PPS ultrafine fiber membrane obtaining the lithium ion battery composite separator (PPSC) with great electrolyte wettability [119], which was conducive to the transfer of lithium ions between electrodes and improve the ionic conductivity [120]. Polyvinylsiloxane (PVS) was used as coating material to prepare PVS/PPS composite separator for lithium ion battery by physical dip coating, and the three-dimensional microporous structure of the composite separator could promote the absorption and storage of electrolyte [121]. The crosslinked polymer electrolyte network was also constructed between PVDF-HFP and hyperbranched polyethyleneimine (PEI) by chemical reaction of C-F bond and amino group on PPS ultrafine fiber membrane surface, which was conducive to the ion transport and the decrease in ohmic polarization degree during battery operation. In addition to battery separator, PPS ultrafine fiber could also be developed and applied in the field of electrolytic cell separator [122]. For example, oxidation, plasma treatment or ultraviolet light irradiation were used to improve the hydrophilicity of PPS fiber, and then made a separator by needling or weaving method for alkaline water electrolytic cells [64,123,124]. Hydrophilic modification was found to be beneficial in improving the water wettability of the separator, and then combined with the high temperature resistance of PPS resin, the performance of the separator in alkaline water electrolytic cells was significantly improved.

#### 5.2.4. Catalytic Field

In recent years, the application potential of PPS ultrafine fiber membrane in the catalysis field has received increasingly more attention because of its large specific surface area and excellent chemical stability, which is particularly conducive to the loading of catalyst and its application in catalytic reaction. Ag_3_PO_4_ nanoparticles were loaded on the sulfonated polyphenylene sulfide (SPPS) ultrafine fiber membrane by precipitation method, and obtained uniform Ag_3_PO_4_/SPPS composites [125], which showed the highest degradation efficiency at 97.8%. And FeC_2_O_4_ was also used to modify PPS catalytic ultrafine fiber membrane (PPS/FeC_2_O_4_) by chemical precipitation method, which was an efficient heterogeneous Fenton catalyst. A possible underlying mechanism of synthetic process is shown in Figure 11, FeC_2_O_4_ crystal grains were firmly anchored on the fiber membrane surface via sulfonated PPS interfacial bonding, which significantly improved the operational stability and reusability of PPS/FeC_2_O_4_ catalyst [126]. However, it is also worth noting that PPS resin is very susceptible to photocatalytic degradation, causing the reduction in mechanical properties. However, was it really suitable for use in the field of photocatalysis? Researchers should pay more attention to how to inhibit the photodegradation of PPS resin in the process of photocatalysis. In addition, sulfonated PPS nonwovens with high specific surface area were prepared and used for the continuous esterification of oleic acid and methanol, showing high-efficient and stable catalytic performance, which showed that the prepared modified PPS material was a promising solid acid catalyst [127].

### 5.3. Development Prospects of PPS Ultrafine Fiber Membrane

PPS melt-blown nonwovens play an irreplaceable role in some fields, such as high-temperature filter membrane, high-efficiency oil absorption material and high-performance lithium–ion battery. Although PPS melt-blown nonwovens have been industrialized in a small scale at home and abroad, there are still many problems, such as the difficulty of continuous production, low uniformity and poor quality. Furthermore, under the premise of giving full play to its unique advantages, it can greatly expand the scope of its application through modification. Researchers should try the following measures to break through this dilemma: (1) Developing fiber grade PPS resin with high linearity and low branched chain. The stable fluidity of the melt can be effectively improved to reduce the spinning difficulty; (2) Developing PPS ultrafine fiber membrane with finer fiber diameter. For example, high-temperature solution electrospinning technology or template powder metallurgy method can be attempted to prepare ultrafine fiber membranes for special industries in harsh environments; (3) Developing new technology for fiber membrane-forming. Melt electrospinning technology has become a mature technology to prepare PPS microfibers, and the PPS fiber membrane can be prepared by improving its receiving device or developing post-processing technology. The wet forming technology can also be developed to rapidly make PPS fiber into PPS fiber membrane; (4) Developing more special applications. PPS ultrafine fiber membrane has high effective permeability, excellent chemical corrosion resistance, thermal stability and electrical insulation, which endow it the potential for application in extreme environment, especially in the separation and purification of high temperature corrosive solid–liquid waste.

## 6. Conclusions and Future Outlook

PPS resin is provided with unique properties, such as excellent chemical stability, great thermal stability and strong solvent resistance; as such, its application in the field of membrane research is increasingly more extensive, especially in harsh environments. In this review, we systematically describe the synthesis methods and research progress of PPS resin and preparation as well as modification methods of PPS membranes, including PPS flat membrane, PPS hollow fiber membrane and PPS ultrafine fiber membrane. The prepared membranes have been applied in many fields, such as oil–water separation, seawater desalination, separation and catalytic degradation of small organic molecules, adsorption, electrolytic cells and battery separators. 

To date, remarkable progress has been made in the research of PPS resin and PPS membrane, which have received significant attention in recent years due to its outstanding properties. However, there are still many defects that restrict the further industrialization or wide range application of various types of PPS membranes, such as poor mechanical properties, low selectivity and permeability and poor membrane operation stability. Therefore, it is still a challenge to produce a desirable PPS membrane with excellent performance.

Future research and development should focus on improvement in membrane performance and development of functionalized PPS membrane. First, researchers need to have a deeper understanding of the basic properties of PPS resin in order to find out the relationship among the material properties, membrane morphology and membrane operation state, which is the most basic part of membrane research. Then, the research direction of membranes is analyzed according to the application field and actual situation of the membrane. Flat membrane is the most basic form of membrane, which is convenient for laboratory preparation and research, and it is very convenient to explore its more advanced preparation and modification methods for stronger membrane application processes. This is contrary to hollow fiber membrane and ultra-fine fiber membrane. The spinning process is complex, but it has the advantage of continuous production, which is suitable for industrial development and is easy to make membrane modules for practical application. The ultimate goal of researchers is to develop industrialization and transform the results of the laboratory into products that can solve practical problems; thus, the development of PPS hollow fiber membrane and ultra-fine fiber membrane has important practical significance. Finally, researchers should give full play to the unique properties of PPS resin and develop more application fields, especially in harsh environments, which not only improve the practical value of PPS membranes, but also solve the problems in production.

## Figures and Tables

**Figure 1 membranes-12-00924-f001:**
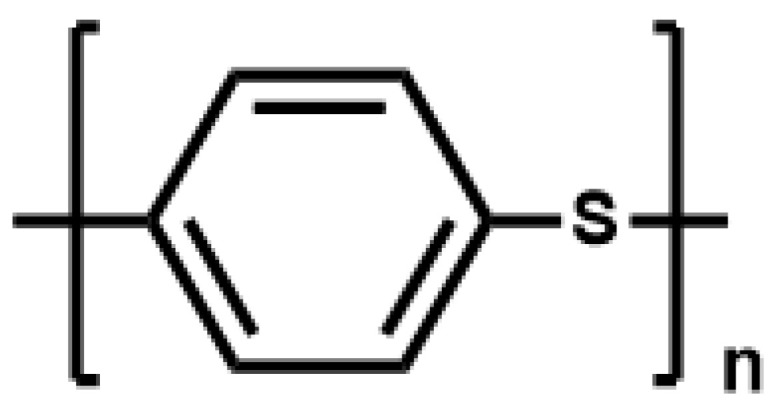
Structure of PPS molecular.

**Figure 2 membranes-12-00924-f002:**
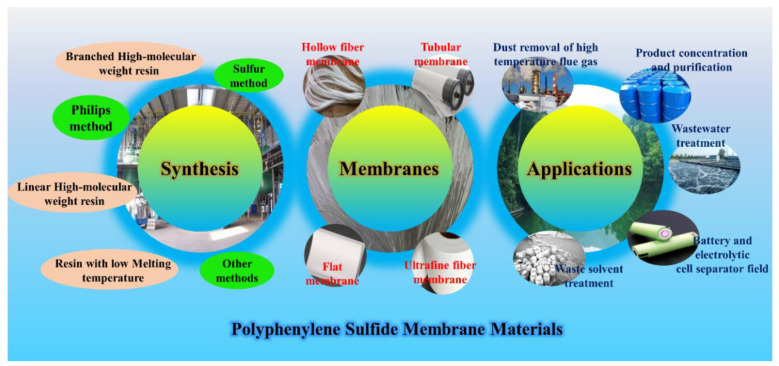
Synthesis methods and application fields of various types of PPS resin.

**Figure 3 membranes-12-00924-f003:**
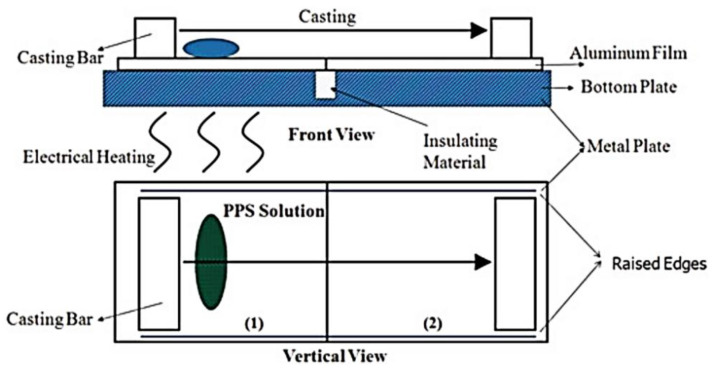
Schematic of the PPS membrane casting device. Reproduced (adapted) with permission from [43], published by Informa UK Ltd., 2014.

**Figure 4 membranes-12-00924-f004:**
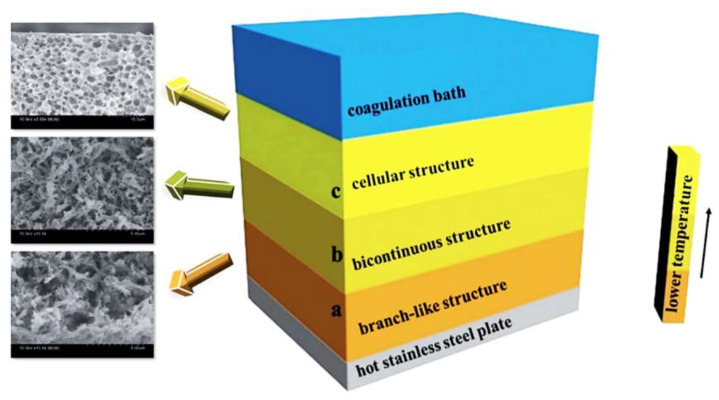
Schematic diagram of phase separation for sandwich-like structure. Reproduced (adapted) with permission from [47], published by Royal Society of Chemistry, 2017.

**Figure 5 membranes-12-00924-f005:**
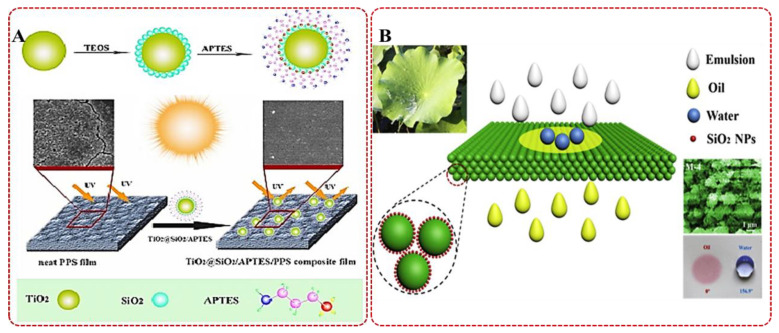
(**A**) The schematic diagram of PPS film modified by functionalized TiO_2_ and its anti-UV performance mechanism. Reproduced (adapted) with permission from [56], published by Royal Society of Chemistry, 2017. (**B**) Oil–water separation process of superhydrophobic-superoleophilic PPS-SiO_2_ hybrid membrane with a lotus leaf-like micro–nano structure. Reproduced (adapted) with permission from [59], published by Elsevier, 2019.

**Figure 6 membranes-12-00924-f006:**
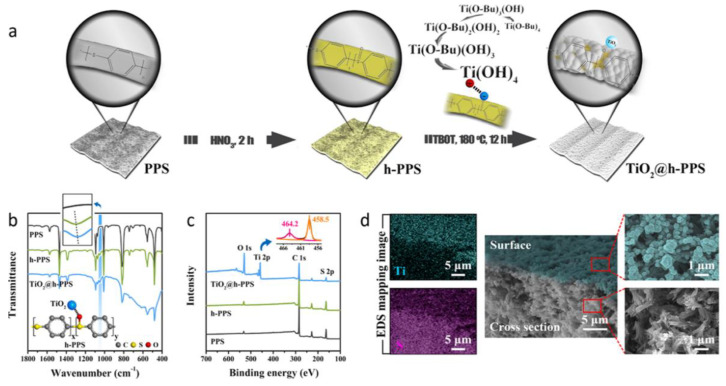
(**a**) Schematic diagram of the fabrication of membranes; (**b**) FT-IR spectra of membranes. (**c**) XPS spectra of membranes; (**d**) SEM images and corresponding Energy Dispersive System (EDS) mapping images of TiO2@h-PPS membrane. Reproduced (adapted) with permission from [69], published by American Chemical Society, 2019.

**Figure 7 membranes-12-00924-f007:**
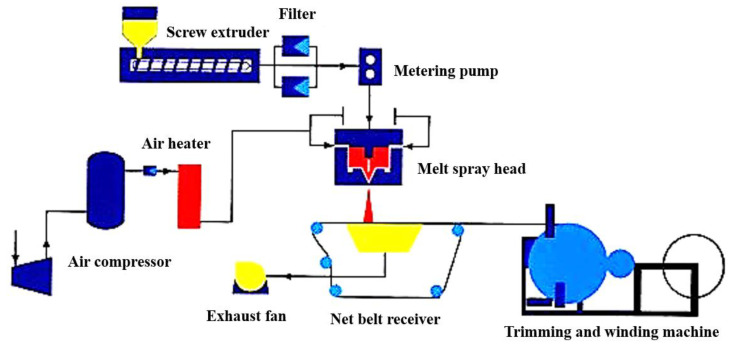
Structure diagram of melt-blown tester.

**Figure 8 membranes-12-00924-f008:**
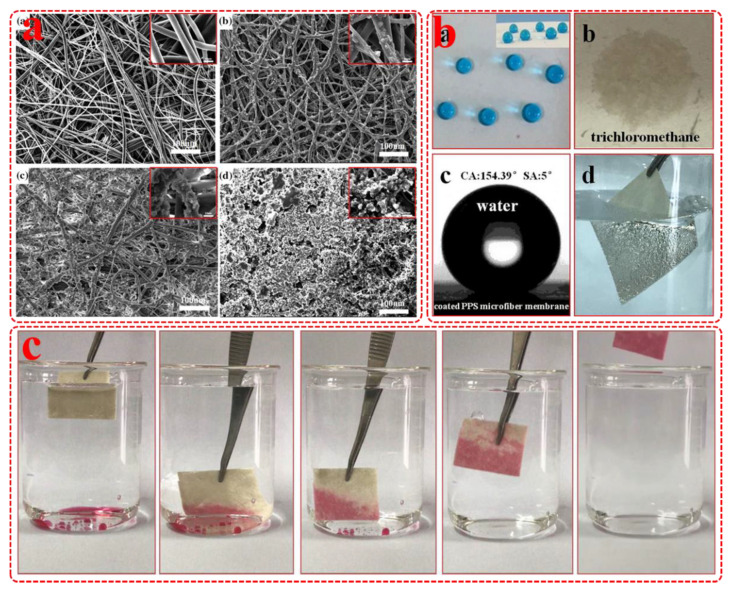
(**a**) SEM images of Pristine and coated PPS ultrafine fiber membranes; (**b**) Morphology photograph of water droplets and oil droplet contact with coated PPS ultrafine fiber membranes; (**c**) Image of trichloromethane removal process under water. Reproduced (adapted) with permission from [106], published by Springer, 2018.

**Figure 9 membranes-12-00924-f009:**
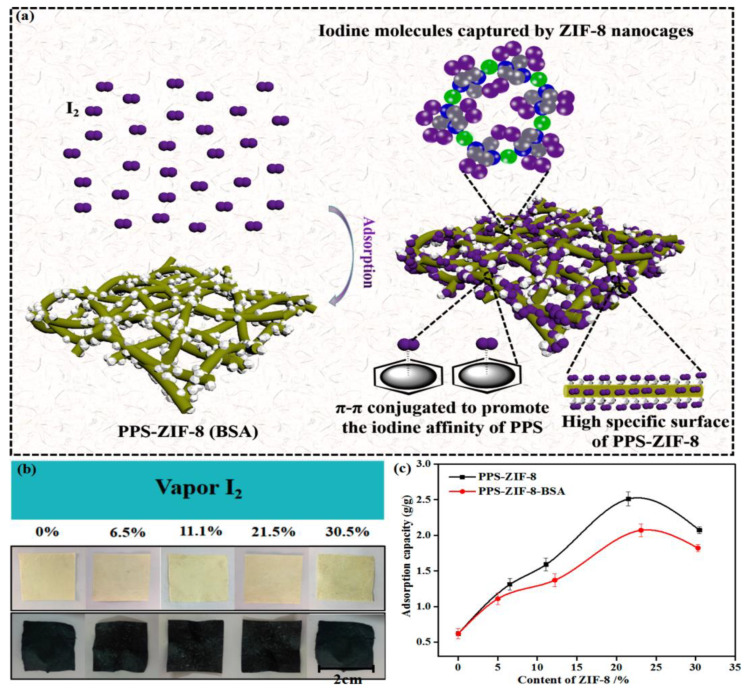
(**a**) Process of iodine vapor capture by the PPS-ZIF-8 (BSA) ultrafine fiber membrane; (**b**) The photograph of PPS-ZIF-8 (BSA) ultrafine fiber membranes containing different ZIF-8 contents before and after iodine absorption; (**c**) Iodine adsorption capacities of PPS-ZIF-8 (BSA) ultrafine fiber membranes. Reproduced (adapted) with permission from [116], published by American Chemical Society, 2019.

**Figure 10 membranes-12-00924-f010:**
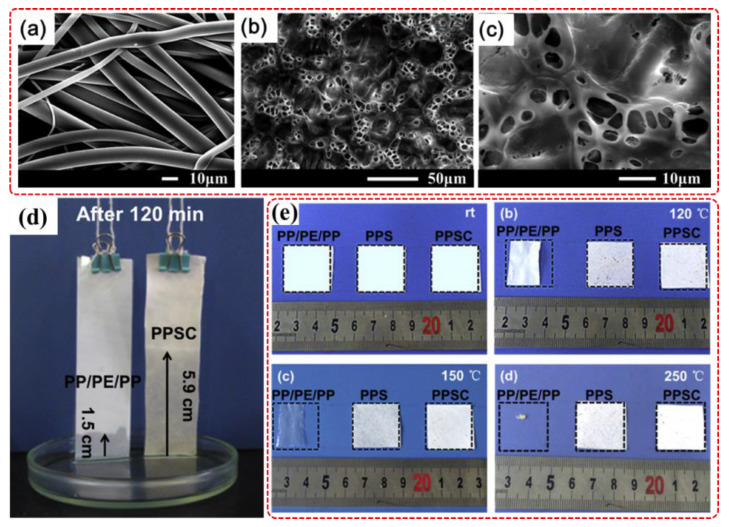
(**a**–**c**) SEM images of PPS ultrafine fiber membrane and PPSC separator; (**d**) Photograph of electrolyte immersion-height of PP/PE/PP separator and PPSC separator after immersion; (**e**) Photographs of PP/PE/PP separator, PPS nonwoven and PPSC separator after thermal treatments at different temperatures. Reproduced (adapted) with permission from [119], published by Elsevier, 2018.

**Figure 11 membranes-12-00924-f011:**
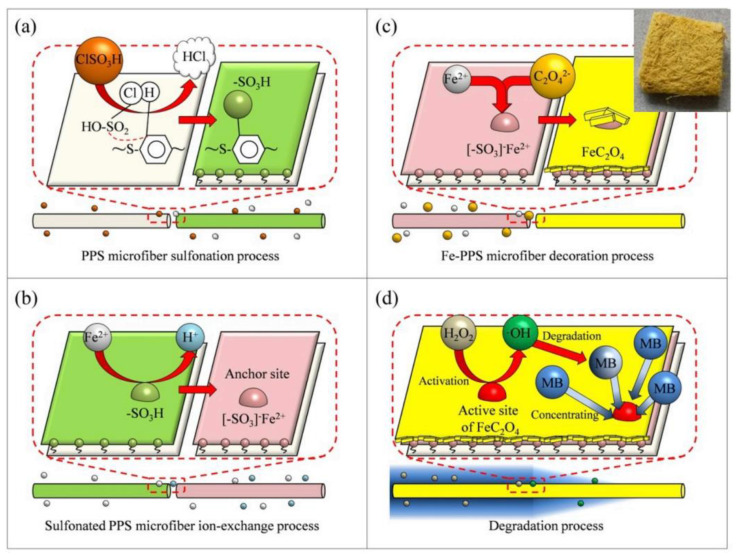
Possible underlying mechanism of synthetic process and performance improvement. Reproduced (adapted) with permission from [126], published by Elsevier, 2019.

## Data Availability

Not applicable.

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
