# Peer review of "Polyphenylene Sulfide-Based Membranes: Recent Progress and Future Perspectives"

_membranes, 2022, doi:10.3390/membranes12100924_

Round 1

Reviewer 1 Report

Polyphenylene Sulfide Based Membranes: Recent Progress and Future Perspective.

This review document intends to develop the state of the art of  Polyphenylene Sulfide Based Membranes. While the topic and specifically this review may be of interest for readers of membranes journal, some comments need to be taken into account before its publication in this journal:

1)    As a major comment: an overwhelming majority of cited works have been prepared by Chinese groups, however, a simple, non-exhaustive research by using Google tool, with only “Polyphenylene Sulfide Membranes” as keywords, give the following documents published by non-Chinese groups and that lack in the review:

M. Viviani, S. P. Fluitman, K. Loos, and G. Portale, Highly Stable Membranes of Poly(phenylene sulfide benzimidazole) Cross-Linked with Polyhedral Oligomeric Silsesquioxanes for High-Temperature Proton Transport, ACS Appl. Energy Mater. 2020, 3, 8, 7873–7884.

R. A. Lundgard, Method for preparing poly(phenylene sulfide) membranes, Patent (expired) US5507984A.

T. J. Fuller and B. T. Dobulis , Sulfonated poly(phenylene sulfide) films as polyelectrolyte membranes, Patent US7601449B2.

J. Hoadley, J. and J. Ginter, Polyphenylene Sulfide (PPS) as a Membrane in Electrolysis Cells, SAE Technical Paper, 1996, 961438.

J. Miyake, R. Taki, T. Mochizuki, R. Shimizu, R. Akiyama, M. Uchida and K. Miyatake, Design of flexible polyphenylene proton-conducting membrane for next-generation fuel cells, Science Advances, 2017, 3, 10, eaao0476.

C. Minsik, L. Jungrok, R. Seongwoo, K. Bon-Cheol, Fabrication and Applications of Polyphenylene Sulfide (PPS) Composites: A Short Review, Composites Research, 2020, 33, 3, 91-100.

M. C. Lopes de Oliveira, I. J. Sayeg, G. Ett, R. Altobelli Antunes, Corrosion behavior of polyphenylene sulfide-carbon black-graphite composites for bipolar plates of polymer electrolyte membrane fuel cells, Int. J. Hydrogen Energy, 2014, 39, 16405-16418.

I strongly suggest to include these references in this review and to do a more exhaustive research concerning this topic to give a complete state of the art on this field.

2)    The sentence in line 470 “At present, the main research focused in China.” is incomplete, however, maybe the authors meant “At present, research about Polyphenylene Sulfide Based Membranes is mainly performed in China”. I suggest to nuance this sentence and if true (no references are given to support this affirmation), authors may give some explanations to this phenomenon.

As minor comments:

3)    The English language needs to be proof checked before publication, for instance, singular/plural concordance and tense sequencing.

4)    Some chemical formulae need to be written in accord to the rules: write Na2S instead of Na2S (line 75)

5)    Line 452, use “cannot” instead of “can’t”

Author Response

Reviewer #1:

This review document intends to develop the state of the art of Polyphenylene Sulfide Based Membranes. While the topic and specifically this review may be of interest for readers of membranes journal, some comments need to be taken into account before its publication in this journal:

  1. As a major comment: an overwhelming majority of cited works have been prepared by Chinese groups, however, a simple, non-exhaustive research by using Google tool, with only “Polyphenylene Sulfide Membranes” as keywords, give the following documents published by non-Chinese groups and that lack in the review:
  2. Viviani, S. P. Fluitman, K. Loos, and G. Portale, Highly Stable Membranes of Poly(phenylene sulfide benzimidazole) Cross-Linked with Polyhedral Oligomeric Silsesquioxanes for High-Temperature Proton Transport, ACS Appl. Energy Mater. 2020, 3, 8, 7873–7884.
  3. A. Lundgard, Method for preparing poly(phenylene sulfide) membranes, Patent (expired) US5507984A.
  4. J. Fuller and B. T. Dobulis, Sulfonated poly(phenylene sulfide) films as polyelectrolyte membranes, Patent US7601449B2.
  5. Hoadley, J. and J. Ginter, Polyphenylene Sulfide (PPS) as a Membrane in Electrolysis Cells, SAE Technical Paper, 1996, 961438.
  6. Miyake, R. Taki, T. Mochizuki, R. Shimizu, R. Akiyama, M. Uchida and K. Miyatake, Design of flexible polyphenylene proton-conducting membrane for next-generation fuel cells, Science Advances, 2017, 3, 10, eaao0476.
  7. Minsik, L. Jungrok, R. Seongwoo, K. Bon-Cheol, Fabrication and Applications of Polyphenylene Sulfide (PPS) Composites: A Short Review, Composites Research, 2020, 33, 3, 91-100.
  8. C. Lopes de Oliveira, I. J. Sayeg, G. Ett, R. Altobelli Antunes, Corrosion behavior of polyphenylene sulfide-carbon black-graphite composites for bipolar plates of polymer electrolyte membrane fuel cells, Int. J. Hydrogen Energy, 2014, 39, 16405-16418.

I strongly suggest to include these references in this review and to do a more exhaustive research concerning this topic to give a complete state of the art on this field.

Response: Thanks for your suggestion. We agree with the reviewer’s comments. And we have cited the references you suggested reasonably and revised the manuscript.

  1. The sentence in line 470 “At present, the main research focused in China.” is incomplete, however, maybe the authors meant “At present, research about Polyphenylene Sulfide Based Membranes is mainly performed in China”. I suggest to nuance this sentence and if true (no references are given to support this affirmation), authors may give some explanations to this phenomenon.

Response: Thank you for your suggestion. But we have not found relevant references to support this affirmation, so we have revised and deleted this sentence to ensure the preciseness of the manuscript.

As minor comments:

  1. The English language needs to be proof checked before publication, for instance, singular/plural concordance and tense sequencing.

Response: We agree with the reviewer’s comments. And we have revised the manuscript language, especially singular/plural concordance and tense sequencing.

  1. Some chemical formulae need to be written in accord to the rules: write Na2S instead of Na2S (line 75)

Response: We agreed with the reviewer’s comments. And we have revised the manuscript, shown in line 75.

  1. Line 452, use “cannot” instead of “can’t”

Response: We agree with the reviewer’s comments. And we have revised the manuscript, shown in line 456.

Reviewer 2 Report

This manuscript reviews different aspects of polyphenylene sulfide membrane application. Manuscript meet the scope of Membranes journal and can be published after minor revision.

Comments are follows:

- PPS synthesis methods should be describes with in more details: difference of structure, molecular weight, yield and typical reaction times. Some details presented for Philips method but not for Sulfur method. It is also desirable to mention few articles for every synthesis method to show variability of synthesis conditions and PPS properties.

- Section 2.4.3. Reasons for such decrease in melting temperature is not explained. Please discuss reasons why obtained resin have lower melting temperature and how it can be achieved.

- Lines 241-250. In [43] authors described quite common devise for membrane preparation with TIPS method. Authors can mention here articles where TIPS method was used for preparing membranes from other polymers, for instance, polypropylene.

- line 616 “In addition, it was reported in the literature that PPS materials...” In this sentence should be links where it was reported.

- Text does not consist link to Figure 2.  

Author Response

Reviewer #2:

This manuscript reviews different aspects of polyphenylene sulfide membrane application. Manuscript meet the scope of Membranes journal and can be published after minor revision.

Comments are follows:

  1. PPS synthesis methods should be described with in more details: difference of structure, molecular weight, yield and typical reaction times. Some details presented for Philips method but not for Sulfur method. It is also desirable to mention few articles for every synthesis method to show variability of synthesis conditions and PPS properties.

Response: Thanks for your suggestion. We agree with the reviewer’s comments. Philips method and Sulfur method are the basic methods to synthesize PPS molecules. In other methods, the methods to synthesize various PPS molecular structures have been introduced in detail.

  1. Section 2.4.3. Reasons for such decrease in melting temperature is not explained. Please discuss reasons why obtained resin have lower melting temperature and how it can be achieved.

Response: Thanks for your suggestion, and we have revised the manuscript, shown in line 175-181.

  1. Lines 241-250. In [43] authors described quite common devise for membrane preparation with TIPS method. Authors can mention here articles where TIPS method was used for preparing membranes from other polymers, for instance, polypropylene.

Response: We agree with the reviewer’s comments. And we have revised the manuscript. And we supplement the references [44-46].

  1. line 616 “In addition, it was reported in the literature that PPS materials...” In this sentence should be links where it was reported.

Response: We agree with the reviewer’s comments. And we have revised the manuscript.

  1. Text does not consist link to Figure 2. 

Response: Thank you for your suggestion, and we have revised the manuscript.

Round 2

Reviewer 1 Report

No other comment on this version.